# Halogen-bonded cocrystallization with phosphorus, arsenic and antimony acceptors

Katarina Lisac [1], Filip Topić [2], Mihails Arhangelskis [2], Sara Cepić [1], Patrick A. Julien [2], Christopher W. Nickels[2], Andrew J. Morris [3], Tomislav Friščić [2] & Dominik Cinčić [1]

The formation of non-covalent directional interactions, such as hydrogen or halogen bonds, is a central concept of materials design, which hinges on using small compact atoms of the 2nd period, notably nitrogen and oxygen, as acceptors. Heavier atoms are much less prominent in that context, and mostly limited to sulfur. Here, we report the experimental observation and theoretical study of halogen bonds to phosphorus, arsenic and antimony in the solid state. Combining 1,3,5-trifluoro-2,4,6-triiodobenzene with triphenylphosphine, -arsine, and -stibine provides cocrystals based on I···P, I···As and I···Sb halogen bonds. The demonstration that increasingly metallic pnictogens form halogen bonds sufficiently strong to enable cocrystal formation is an advance in supramolecular chemistry which opens up opportunities in materials science, as shown by colossal thermal expansion of the cocrystal involving I···Sb halogen bonds.

[1] Faculty of Science, Department of Chemistry, University of Zagreb, Horvatovac 102a, HR-10000 Zagreb, Croatia. [2] Department of Chemistry, McGill University, 801 Sherbrooke St. W, Montreal H3A 0B8, Canada. [3] School of Metallurgy and Materials, University of Birmingham, Edgbaston, Birmingham B15 2TT, UK. Correspondence and requests for materials should be addressed to T.Fščić. (email: tomislav.friscic@mcgill.ca) or to D.Cčić. (email: dominik@chem.pmf.hr)

Halogen bonding (XB), an attractive interaction between an electrophilic region on a halogen atom and a nucleophile[1], has emerged as one of the most important directional intermolecular forces, relevant for the design of functional solids, separations, pharmaceuticals, biomolecular recognition and more[2–6]. Compared to hydrogen bonding, XB is expected to provide access to molecular assembly motifs involving heavy atoms with increasingly diffuse electron orbitals, that would engage in hydrogen bonding only with difficulty or not at all[7]. This is shown in reports of XB cocrystals involving later members of group 16, e.g., S, Se, and, most recently, Te[8–12]. In contrast, until recently[13] there have been no reports of analogous cocrystals of group 15 elements (the pnictogens) except nitrogen[14–16], and documented examples of XB interactions to heavier pnictogens appear limited to gas-phase studies.

Here, we report XB cocrystals involving neutral phosphorus, arsenic and antimony acceptors, consisting of 1,3,5-trifluoro-2,4,6-triiodobenzene (**tftib**)[17–20] as the donor and triphenylphosphine (**PPh₃**), -arsine (**AsPh₃**), or -stibine (**SbPh₃**) as acceptors (Fig. 1a). The cocrystals can be made from solution or in the solid state and can be recrystallized without forming halonium salts.

## Results

**Synthesis and structures of cocrystals.** The cocrystal (**tftib**)(**PPh₃**) was obtained as a part of our program in exploring XB to unusual acceptors[21]. Slow evaporation of an equimolar solution of **tftib** and **PPh₃** in acetonitrile gave colorless crystals that were characterized by single crystal X-ray diffraction at 103 K. Structure determination revealed that asymmetric unit consists of one molecule each of **tftib** and **PPh₃** (Fig. 1b). One iodine atom in each **tftib** unit is engaged in a short I···P contact to a **PPh₃** molecule, with the interatomic distance $d_{XB,exp}$ of 3.3133(5) Å and $\angle_{C-I\cdots P}$ angle of 165.33(4)° (Table 1, full crystallographic data is provided in Supplementary Table 1). The I···P separation is 12.3% shorter ($R_{XB} = 0.877$)[22] than the sum of van der Waals radii of phosphorus and iodine[23], consistent with XB. It is also significantly longer than the covalent P–I bond formed by reaction of **PPh₃** and $I_2$ (2.48 Å, which corresponds well to 2.46(4) Å, the

sum of covalent radii for P and I, 1.07(3) and 1.39(3) Å[24], respectively)[25]. The remaining iodine atoms on each **tftib** molecule form a short I···F contact of 3.1011(10) Å with a neighboring **tftib** molecule, and an I···π contact with another neighboring **PPh₃** unit, with I···C distances of 3.3456(16), 3.5642(16), and 3.6592(18) Å (Fig. 2, Supplementary Fig. 1). The **PPh₃** molecules in the crystal structure of (**tftib**)(**PPh₃**) are arranged in pairs situated around a center of inversion, forming the phenyl embrace motif (Supplementary Fig. 2) that has been extensively investigated by the Dance group[26]. While this work was under review, the cocrystal (**tftib**)(**PPh₃**) was also observed by the Bryce group, and characterized by single crystal X-ray diffraction and solid-state nuclear magnetic resonance (NMR) spectroscopy[13].

Next, we explored cocrystallization of **tftib** with **AsPh₃**, **SbPh₃**, triphenylbismuth (**BiPh₃**) and triphenylamine (**NPh₃**). Crystallization from acetonitrile gave colorless crystals of (**tftib**)(**AsPh₃**) and (**tftib**)(**SbPh₃**) which were analyzed by single crystal X-ray diffraction at 103 K. Both cocrystals were isostructural to (**tftib**)(**PPh₃**) (Table 1, full crystallographic data is given in Supplementary Tables 2 and 3, also Supplementary Figs. 1, 2), exhibiting

**Table 1 Crystallographic parameters, XB distances, angles, $R_{XB}$ and $R_{I>2\sigma}$ for (tftib)(PPh₃), (tftib)(AsPh₃) and (tftib)(SbPh₃), determined at 103 K**

|  | (tftib)(PPh₃) | (tftib)(AsPh₃) | (tftib)(SbPh₃) |
|---|---|---|---|
| $a$ (Å) | 9.1186(8) | 9.1477(8) | 9.1863(10) |
| $b$ (Å) | 10.8976(10) | 11.0051(9) | 11.2726(12) |
| $c$ (Å) | 13.3301(12) | 13.3755(12) | 13.3561(14) |
| $\alpha$ (°) | 88.147(3) | 88.515(3) | 89.190(4) |
| $\beta$ (°) | 72.738(3) | 71.685(3) | 70.066(4) |
| $\gamma$ (°) | 76.213(3) | 76.816(3) | 78.455(4) |
| $d_{XB,exp}$ (Å) | 3.3133(5) | 3.4211(3) | 3.5747(3) |
| $R_{XB}$ | 0.877 | 0.893 | 0.885 |
| $\angle_{C-I\cdots E}$ (°) | 165.33(4) | 166.02(5) | 168.28(6) |
| $R_{I>2\sigma}$ (%) | 2.66 | 3.13 | 3.12 |

Note: $R_{XB} = d(X\cdots Y)/(r_X + r_Y)$; $d(X\cdots Y)$ is the distance between X and Y in an R−X···Y halogen bond; $r_X$ and $r_Y$ are the respective vdW radii of X and Y[22,23]

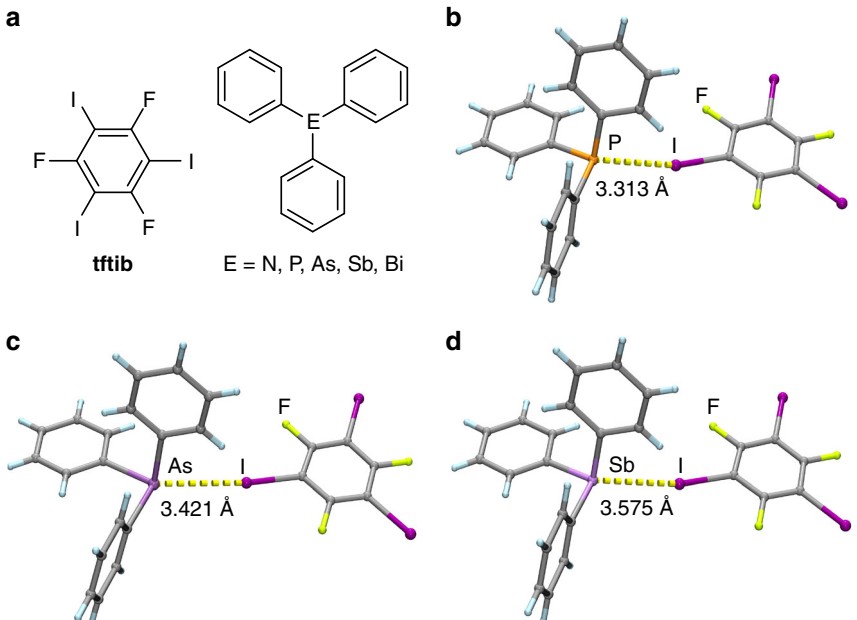

**Fig. 1** Cocrystals with I···P, I···As, and I···Sb halogen bonds. **a** Schematic view of the herein explored halogen bond donor **tftib** and the acceptors **NPh₃**, **PPh₃**, **AsPh₃**, **SbPh₃**, and **BiPh₃**. Structures of the halogen-bonded molecular assemblies in the cocrystals: **b** (**tftib**)(**PPh₃**), **c** (**tftib**)(**AsPh₃**), and **d** (**tftib**)(**SbPh₃**)

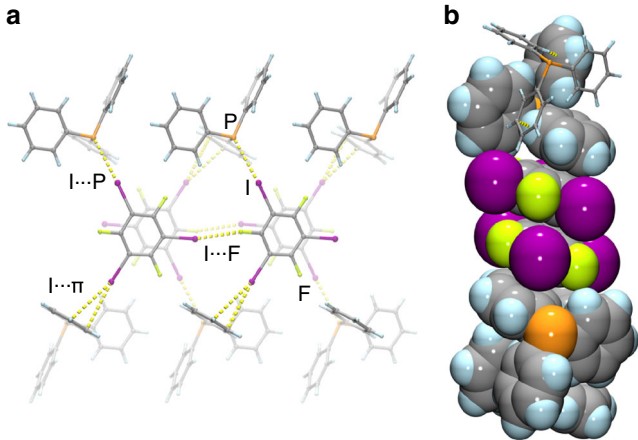

**Fig. 2** Overview of molecular packing in the cocrystal (**tftib**)(**PPh₃**). **a** Each of the iodine atoms of the per **tftib** molecule is involved in a short I⋯P interaction with a neighboring **PPh₃** unit, an I⋯π contact with another neighboring **PPh₃** molecule or a short I⋯F contact to a neighboring **tftib** molecule. **b** Pairs of **tftib** molecules in the crystal structure stack with phenyl rings of two neighboring **PPh₃** moieties. The cocrystal structures of (**tftib**)(**AsPh₃**) and (**tftib**)(**SbPh₃**) are isostructural and corresponding crystal structure views are shown in the Supplementary Fig. 1

short I⋯As and I⋯Sb contacts of 3.4211(3) Å (Fig. 1c) and 3.5747 (3) Å (Fig. 1d), respectively. Both contacts are highly linear and ca. 11% shorter than respective sums of van der Waals radii[23], consistent with XB. Notably, the I⋯As separation in (**tftib**) (**AsPh₃**) is significantly longer than the As-I covalent bond in the adduct **AsPh₃I₂** (2.64 Å, which corresponds well to 2.58(5) Å, the sum of covalent radii for As and I, 1.19(4) and 1.39(3) Å[24], respectively)[25,27].

**Mechanochemical synthesis.** The cocrystals (**tftib**)(**PPh₃**), (**tftib**) (**AsPh₃**), and (**tftib**)(**SbPh₃**) were also accessible mechanochemically[28], by ball milling neat or by liquid-assisted grinding (LAG) in the presence of a small amount of ethanol (liquid−to −solid ratio, $\eta = 0.2\ \mu L\ mg^{-1}$)[29]. Milling equimolar amounts of **tftib** and the XB acceptor gave solid materials whose powder X-ray diffraction (PXRD) patterns matched the simulated ones for (**tftib**)(**PPh₃**), (**tftib**)(**AsPh₃**), and (**tftib**)(**SbPh₃**) (Fig. 3a, also Supplementary Figs. 3-5). All attempts to synthesize cocrystals (**tftib**)(**BiPh₃**) and (**tftib**)(**NPh₃**) mechanochemically, or from solution, have been unsuccessful (see Supplementary Figs. 6, 7).

The cocrystals and reaction mixtures were also analyzed by Fourier-transform infrared attenuated total reflectance (FTIR-ATR, see Supplementary Fig. 8) and Raman spectroscopy. In particular, mechanochemical cocrystallization was also followed in situ by Raman spectroscopy, especially by monitoring the low intensity bands of solid reactants and cocrystals between 1100 and 1500 cm⁻¹. Such experiments, which required the use of sapphire milling jars that are considerably more transparent in that spectral region compared to more conventional PMMA ones[30], revealed that the cocrystals form rapidly, in less than 10 min by neat milling (Fig. 3b, c, d, for full spectra see Supplementary Figs. 9-11). The reactions by LAG were even faster, and the synthesis of (**tftib**)(**PPh₃**) was accomplished in less than 2 min in the presence of a small amount of ethanol ($\eta = 0.2\ \mu L\ mg^{-1}$) (see Supplementary Figs. 12, 13).

The formation of halogen-bonded cocrystals between **tftib** donor and **PPh₃**, **AsPh₃**, or **SbPh₃** contrasts the documented behavior of organophosphines and -arsines towards strong XB donor halogens and interhalogens, which usually results in covalent bonding between pnictogens and halogens[25,27].

Importantly, whereas the resulting compounds readily dissociate into halonium salts in solution, the herein prepared cocrystals can be readily recrystallized from different solvents, demonstrating reversible XB assembly.

**Theoretical calculations and modeling.** To further evaluate the I⋯P, I⋯As, and I⋯Sb halogen bonds, as well as to explain the inability to synthesize cocrystals of **BiPh₃** and **NPh₃**, we conducted a theoretical density functional theory (DFT) study of putative gas-phase complexes (**tftib**)(**PPh₃**), (**tftib**)(**AsPh₃**), (**tftib**)(**SbPh₃**), (**tftib**)(**BiPh₃**), and (**tftib**)(**NPh₃**), as well as of the crystal structures of (**tftib**)(**PPh₃**), (**tftib**)(**AsPh₃**), (**tftib**)(**SbPh₃**) and the hypothetical crystal structure of (**tftib**)(**BiPh₃**) using periodic DFT. The geometries of optimized solid-state and gas-phase structures are provided in Supplementary Data 1–14 (see Supplementary Notes 1 and 2).

Computational modeling of halogen bonding interactions is an active area of research, with particular effort directed towards understanding the nature of halogen bonding through various energy decomposition schemes[31,32]. Benchmarks of DFT functionals against higher level Coupled-Cluster with Single and Double and Perturbative Triple excitations (CCSD(T)) and second order Møller-Plesset perturbation theory (MP2) calculations have shown that range-separated functionals perform well for modeling halogen bonds to delocalized π-systems[33]. Similarly, long-range corrected functionals have shown superior performance to typical oxygen- and nitrogen-based acceptors[34–36]. Here, we used the range-separated ωB97X functional[35] which is expected to yield a more accurate description of XB contacts and crystal structure as a whole[36]. We have also decided not to use semi-empirical dispersion corrections (SEDCs), as there is no consensus whether they offer an improved treatment of halogen bonds[32,36].

Periodic and gas-phase geometry optimizations were performed using CRYSTAL17[37] with all-electron basis sets for H, C, F, N, P, and As[38]. Effective core potentials (ECPs)[39] were used to account for relativistic effects in heavy atoms (Sb, I[40], Bi[41]). Cocrystal formation energies ($\Delta E_{f,calc}$, Table 2 and Supplementary Table 4) were calculated from lattice energies of optimized crystal structures, including the unit cell dimensions, of the three synthesized cocrystals, the putative cocrystal (**tftib**)(**BiPh₃**), and structures of reactants found in the Cambridge Structural Database (CSD, v. 5.39, Feb 2018)[42]. Supplementary Data 1–9 contain all DFT-optimized solid-state structures in CIF format (see Supplementary Note 1). Energies of isolated XB dimers were computed for molecular geometries found in cocrystals, and for optimized gas-phase geometries (see Supplementary Tables 5-7). Supplementary Data 10–14 contain geometries for all optimized gas-phase dimers in XYZ format (see Supplementary Note 2). In order to rationalize the varying halogen-bonding abilities of the pnictogen atoms, electrostatic potential surfaces (ESPs, Fig. 4a, Table 2) were computed for isolated molecules in Gaussian 16[43,44].

In order to assess the suitability of our chosen DFT functional, we have compared the geometries of DFT-optimized and experimental cocrystal structures. The differences between calculated and measured halogen bond lengths all fall within 0.1 Å (see Supplementary Table 8), which is highly satisfactory and accounts for less than 3% relative error. The errors in calculated halogen bond angles (∠C−I⋯E) are also highly satisfactory and fall within 6°, or 4% of their experimental values (see Supplementary Table 8). These results are all in line with the general accuracy of non-covalent interactions achievable with periodic DFT methods[45]. Whereas a benchmark theory vs. experiment study for the accuracy of halogen bond geometries

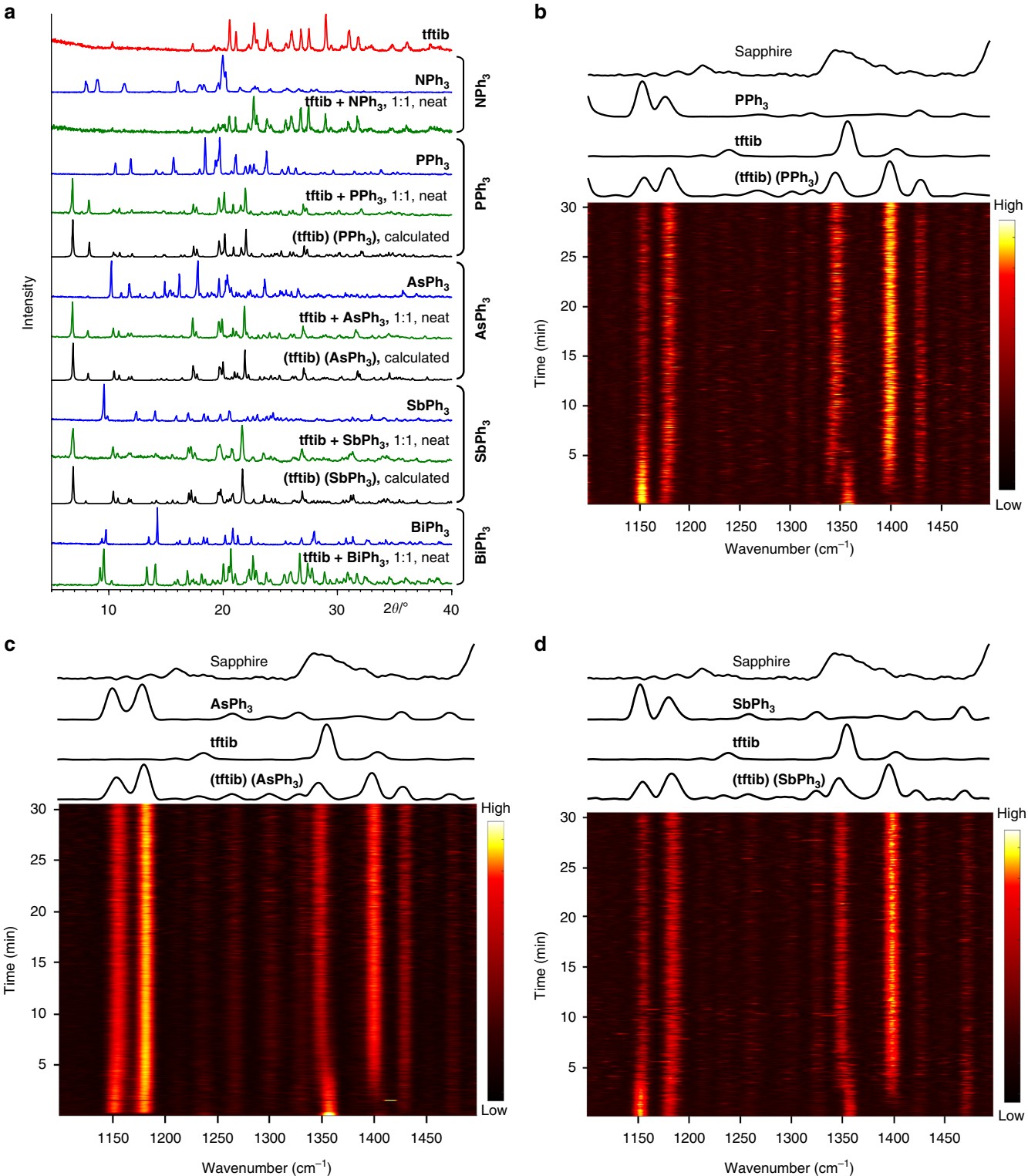

**Fig. 3** Mechanochemical synthesis of the cocrystals. **a** Comparison of powder X-ray diffraction (PXRD) patterns relevant for the observed mechanochemical cocrystallization reactions of **tftib** with **PPh₃**, **AsPh₃**, and **SbPh₃**, as well as attempted mechanochemical syntheses of (**tftib**)(**NPh₃**) and (**tftib**)(**BiPh₃**) by neat milling. Results of corresponding LAG experiments are shown in the Supplementary Figs. 3-7. Real-time collected time-resolved Raman spectra for the mechanochemical neat milling reactions of **tftib** with: **b** PPh₃, **c** AsPh₃, and **d** SbPh₃, demonstrating the rapid formation of the corresponding cocrystals

calculated using periodic DFT has not yet been reported, we note that the herein match of theory and experiment is significantly improved compared to a recent comparison of halogen bond geometries calculated in the gas phase and measured in a crystal

structure[46]. The accuracy of our modeling approach was also verified by comparing the experimentally measured (at 103 K) and DFT-optimized unit cell parameters (see Supplementary Table 9)[47,48].

**Table 2 Calculated cocrystal formation energies ($\Delta E_{f,calc}$), halogen bond energies ($E_{XB,calc}$), distances ($d_{XB,calc}$) and ESP minima on the pnictogen atom or the phenyl ring for (tftib)(PPh$_3$), (tftib)(AsPh$_3$), (tftib)(SbPh$_3$), (tftib)(BiPh$_3$)$^a$, and (tftib)(NPh$_3$)$^b$ in cocrystal and gas phase**

| acceptor | $\Delta E_{f,calc}$ (kJ mol$^{-1}$) | $E_{XB,calc}$ (crystal, kJ mol$^{-1}$) | $E_{XB,calc}$ (gas, kJ mol$^{-1}$) | $d_{XB,calc}$ (crystal, Å) | $d_{XB,calc}$ (gas, Å) | ESP (pnictogen, kJ mol$^{-1}$) | ESP (phenyl, kJ mol$^{-1}$) |
|---|---|---|---|---|---|---|---|
| NPh$_3$ | _$^b$ | _$^b$ | 13.87 | _$^b$ | 3.149 | −28.9$^c$ | −64.6 |
| PPh$_3$ | −15.40 | 16.96 | 22.79 | 3.363 | 3.318 | −118.4 | −62.0 |
| AsPh$_3$ | −11.38 | 13.12 | 15.66 | 3.529 | 3.514 | −95.8 | −63.5 |
| SbPh$_3$ | −7.36 | 12.88 | 15.67 | 3.577 | 3.589 | −89.8 | −61.4 |
| BiPh$_3^a$ | +1.27 | 5.65 | 7.23 | 3.803 | 3.817 | −6.0 | −66.4 |

$^a$Putative structure for (tftib)(BiPh$_3$) cocrystal was generated from that of (tftib)(SbPh$_3$) by replacing the Sb atom with Bi and optimization
$^b$Different geometry of NPh$_3$ compared to its congeners prevented generating a structure for the putative (tftib)(NPh$_3$) cocrystal
$^c$Value corresponds to the point in the center of the molecule closest to the N atom

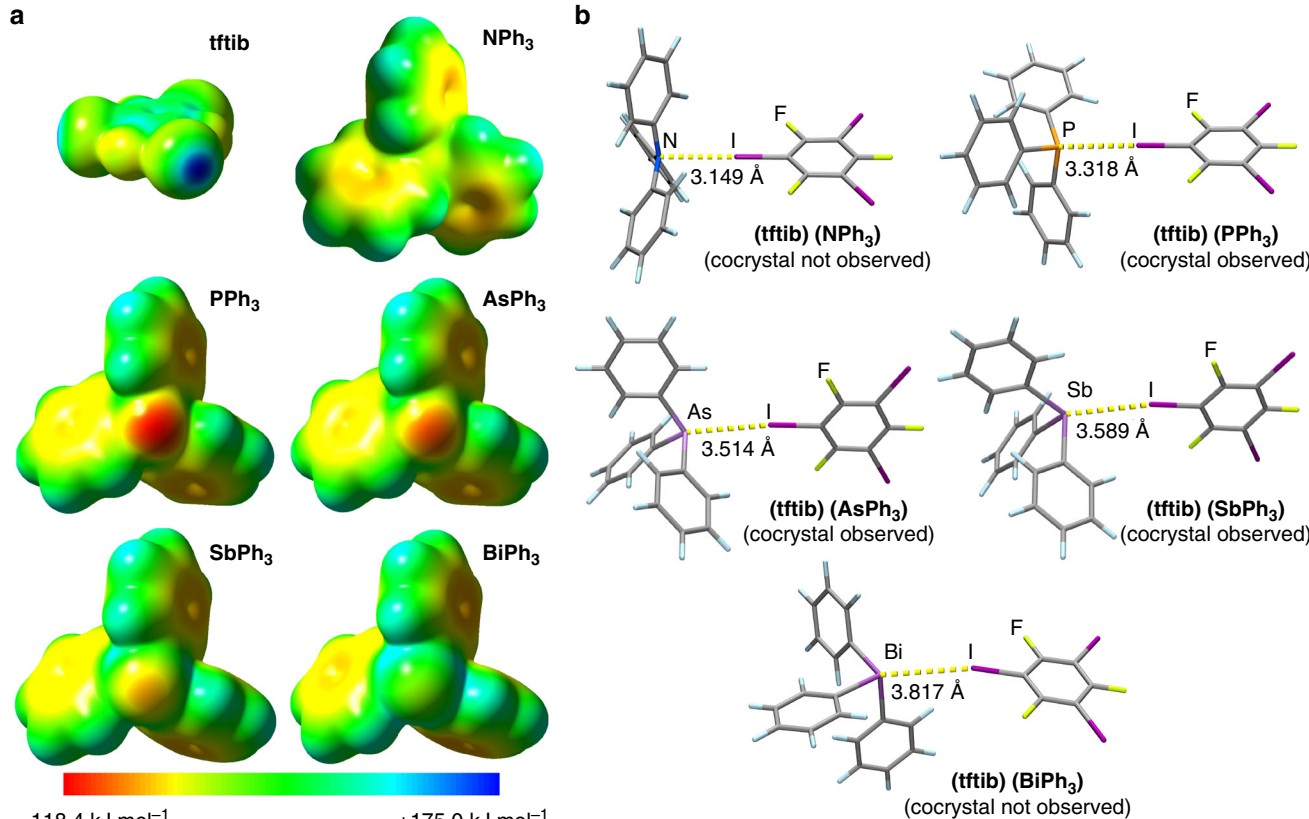

**Fig. 4** Theoretical analysis of halogen-bonded complexes. **a** Comparison of calculated ESPs for the donor and all acceptors with isosurfaces plotted at 0.002 a.u. **b** Optimized gas-phase geometries for the halogen-bonded complexes (**tftib**)(**NPh$_3$**), (**tftib**)(**PPh$_3$**), (**tftib**)(**AsPh$_3$**), (**tftib**)(**SbPh$_3$**), and (**tftib**)(**BiPh$_3$**). Supplementary Data 10–14 contain geometries for all optimized gas-phase dimers in XYZ format (see Supplementary Note 2)

Calculated halogen bond dimer energies ($E_{XB,calc}$) in the crystal and in the gas phase follow the trend **PPh$_3$** > **AsPh$_3$** > **SbPh$_3$** > **NPh$_3$** (calculated in gas phase only) > **BiPh$_3$**. This trend is explained by a combination of at least three factors. The first one, pertinent to P, As, Sb, and Bi, is an expected drop in the ESP of the pyramidally-bonded pnictogen atom due to decreasing electronegativity, in the order **PPh$_3$** > **AsPh$_3$** > **SbPh$_3$** > **BiPh$_3$**. The two other factors explain the low energy of halogen-bonded complex formation with **NPh$_3$**: the very low ESP on the nitrogen atom due to delocalization of the lone electron pair with the phenyl substituents[49], and the significantly higher steric bulk that is evident on the nitrogen atom due to the planar arrangement of the three C–N bonds (Fig. 4). The role of steric bulk is particularly notable, considering that

sterically less hindered dimethylaniline derivatives are known to form halogen-bonded cocrystals[50]. Importantly, for **PPh$_3$**, **AsPh$_3$**, and **SbPh$_3$**, the ESP on the pnictogen is more negative than on the phenyl ring π-system, while the opposite is true for **NPh$_3$** and **BiPh$_3$**, which did not form cocrystals (Table 2). An additional consideration for halogen-bonded complex with **BiPh$_3$** concerns the effect of spin-orbit coupling (SOC) interactions on halogen bond energy. Previous computational studies[51,52] have shown that SOC may have a pronounced effect on the energies of halogen bonds to heavy atoms. Our calculation using ADF2018[53,54], however, has shown that SOC accounts for only a 0.1 kJ mol$^{-1}$ change in (**tftib**)(**BiPh$_3$**) dimer interaction energy, compared to non-relativistic DFT calculation (Supplementary Table 6).

The $E_{XB,calc}$ for (**tftib**)(**PPh$_3$**) in the gas phase is similar to that previously calculated[55] for gas-phase XB complex of P(CH$_3$)$_3$ and CF$_3$I (5.76 kcal mol$^{-1}$, 24.10 kJ mol$^{-1}$). However, the $E_{XB,calc}$ for gas-phase (**tftib**)(**NPh$_3$**) is lower than previously calculated for the CF$_3$I···N(CH$_3$)$_3$ complex (5.64 kcal mol$^{-1}$, 23.56 kJ mol$^{-1}$), which can again be explained by planar geometry and delocalization in **NPh$_3$** in contrast to pyramidal trimethylamine that was used in the previous theoretical study. Specifically, the geometry around the nitrogen atom in the hypothetical gas-phase complex (**tftib**)(**NPh$_3$**) remains strongly planar, with the nitrogen atom calculated to be only 0.19 Å out of the plane of the three directly connected carbon atoms. For comparison, the corresponding calculated out-of-plane distances for gas-phase complexes involving the pyramidal molecules **PPh$_3$**, **AsPh$_3$**, **SbPh$_3$**, and **BiPh$_3$** are respectively 0.80, 0.91, 1.09, and 1.19 Å (see Supplementary Data 10–14 for geometries of all optimized gas-phase dimers in XYZ format, Supplementary Note 2).

Periodic DFT calculations were used to evaluate the relative stabilities of synthesized (**tftib**)(**PPh$_3$**), (**tftib**)(**AsPh$_3$**), and (**tftib**)(**SbPh$_3$**) cocrystals. The calculations were verified by comparing the XB lengths ($d_{XB,calc}$) in optimized crystal structures to measured ones. As the molecular geometry of **BiPh$_3$** strongly resembles that of **AsPh$_3$** and **SbPh$_3$**, we also evaluated the structure of a putative cocrystal (**tftib**)(**BiPh$_3$**), which was generated by replacing the Sb atom in the crystal structure of (**tftib**)(**SbPh$_3$**) with Bi and optimizing. For **NPh$_3$**, which exhibits a planar geometry notably different from that of its congeners, this was not possible without attempting a full-scale crystal structure prediction study, which would be outside of the scope of this work. Consequently, we did not investigate any putative structures for a (**tftib**)(**NPh$_3$**) cocrystal. Supplementary Data 1–9 contain all DFT-optimized solid-state structures in CIF format (see Supplementary Note 1).

Gratifyingly, the calculated formation energies ($\Delta E_{f,calc}$) were found to be negative for all of the observed cocrystals, indicating thermodynamic stability with respect to solid reactants. At the same time, the $\Delta E_{f,calc}$ was slightly positive for the so far inaccessible (**tftib**)(**BiPh$_3$**), indicating that the formation of that cocrystal would not be likely (Table 2). Notably, the $\Delta E_{f,calc}$ is most negative for (**tftib**)(**PPh$_3$**) and becomes less negative going down group 15, in agreement with the trend in calculated halogen bond energies and ESPs. Consequently, theoretical calculations conducted so far are consistent with the results of experimental cocrystal screening. As **BiPh$_3$** is of similar molecular shape and dimensions to **PPh$_3$**, **AsPh$_3$**, and **SbPh$_3$**, such agreement between experimental and theoretical results indicates that the I···P, I···As, and I···Sb halogen bonds are an important contributor to enabling the formation of herein observed cocrystals.

**Stability and thermal properties of cocrystals.** Isostructurality of (**tftib**)(**PPh$_3$**), (**tftib**)(**AsPh$_3$**), and (**tftib**)(**SbPh$_3$**) provides an opportunity to compare how switching between I···P, I···As, and I···Sb halogen bonds may affect solid-state properties of materials. For that purpose, we have explored the thermal behavior of the three cocrystals (see Supplementary Figs. 14-26) and investigated their structures at different temperatures (Tables 3, 4, also Supplementary Tables 1-3). Differential scanning calorimetry (DSC) reveals the melting points ($T_m$) of (**tftib**)(**PPh$_3$**), (**tftib**)(**AsPh$_3$**), (**tftib**)(**SbPh$_3$**) as 106, 92, and 77 °C, with associated enthalpies of fusion ($\Delta H_{fus}$) of 31.3, 29.8, and 21.2 kJ mol$^{-1}$. The relative order of $T_m$ and $\Delta H_{fus}$ suggests the ordering of XB strengths I···P > I···As > I···Sb, similar to the calculated one: I···P > I···As ≈ I···Sb.

In order to obtain further insight into the behavior of I···P, I···As, and I···Sb halogen bonds in the solid state, and explore their potential effect on cocrystal properties, we conducted the

**Table 3 Melting points ($T_m$) and the enthalpies of fusion ($\Delta H_{fus}$) for cocrystal components tftib, PPh$_3$, AsPh$_3$ and SbPh$_3$, and the cocrystals (tftib)(PPh$_3$), (tftib)(AsPh$_3$), and (tftib)(SbPh$_3$)**

|  | $T_m$ (°C) | $\Delta H_{fus}$ (kJ mol$^{-1}$) |
|---|---|---|
| **tftib** | 152.90 | 16.15 |
| **PPh$_3$** | 78.79 | 14.49 |
| **AsPh$_3$** | 59.23 | 12.40 |
| **SbPh$_3$** | 52.84 | 13.13 |
| (**tftib**)(**PPh$_3$**) | 106.23 | 31.26 |
| (**tftib**)(**AsPh$_3$**) | 92.10 | 29.84 |
| (**tftib**)(**SbPh$_3$**) | 77.06 | 21.18 |

crystallographic characterization at 103, 153, 203, and 253 K (Table 4, also Supplementary Tables 1-3). The analysis of the crystal structures collected at different temperatures reveals a significant effect of switching between **PPh$_3$**, **AsPh$_3$**, and **SbPh$_3$** on thermal expansion properties of the cocrystals.

Linear expansion coefficients along the principal axes calculated through PASCal[56], using the cocrystal unit cell parameters revealed a colossal positive thermal expansion[57,58] of 121(4) MK$^{-1}$ along one of the axes for (**tftib**)(**SbPh$_3$**) (Fig. 5a).

For (**tftib**)(**AsPh$_3$**) and (**tftib**)(**PPh$_3$**), respective maximal linear expansions coefficients are 89(3) and 83.5(19) MK$^{-1}$ (see Supplementary Figs. 27, 28). Structural analysis at different temperatures also shows that I···P, I···As and I···Sb distances change in a roughly linear fashion, with thermal expansion coefficients of 3.6(2)·10$^2$, 4.5(2)·10$^2$, and 3.9(3)·10$^2$ Å MK$^{-1}$, respectively (Fig. 5b). While these do not directly reflect XB strength[59], they are larger than those reported for I···N (2.8·10$^2$ Å MK$^{-1}$) and Br···N (3·10$^2$ Å MK$^{-1}$) interactions, which is consistent with halogen bonds to P, As and Sb being weaker.

## Discussion

The present study reports the observation of halogen bonds involving heavy pnictogens as acceptors in the solid state. The herein presented cocrystal structures, combined with gas-phase and solid-state theoretical calculations, show that P, As, and Sb can act as respectable XB acceptors in the solid state, forming contacts that are 11–12% shorter than the sums of van der Waals radii, with calculated dissociation energies between 12–17 kJ mol$^{-1}$. Importantly, the described cocrystals can be readily obtained and even re-crystallized from organic solvents, without forming halonium salts. We speculate that the use of an organic halogen bond donor 1,3,5-trifluoro-2,4,6-triiodobenzene, is key to the formation of stable cocrystals, contrasting the covalent bonds and halonium salts that are traditionally observed with strong inorganic donors like halogens and interhalogens. Consequently, it is likely that further development of halogen-bonded supramolecular chemistry of heavy pnictogens will require careful design and manipulation of halogen bond donor strength. While demonstrating yet unexplored supramolecular chemistry of metalloids As and Sb, which has so far been limited to acting as donor atoms in coordination complexes[60], this work also creates opportunities for the design of molecular solids through directional, non-covalent interactions involving heavy, increasingly metallic acceptor atoms.

## Methods

**General information.** All solvents used for syntheses and crystal growth were of reagent grade and were used as received. Triphenylamine (**NPh$_3$**), triphenylphosphine (**PPh$_3$**), triphenylarsine (**AsPh$_3$**), triphenylstibine (**SbPh$_3$**), and

**Table 4 Geometrical parameters of C−I···E (E=P, As, Sb) halogen bonds at different temperatures**

| Cocrystal | (tftib)(PPh$_3$) @103 K | (tftib)(PPh$_3$) @153 K | (tftib)(PPh$_3$) @203 K | (tftib)(PPh$_3$) @253 K |
|---|---|---|---|---|
| $T$ (K) | 103.0(1) | 153.0(1) | 203.0(1) | 253.0(1) |
| $d_{XB,exp}$ (Å) | 3.3133(5) | 3.3280(5) | 3.3469(6) | 3.3669(8) |
| $R_{XB}$ | 0.877 | 0.880 | 0.885 | 0.891 |
| $\angle_{C-I···E}$ (°) | 165.33(4) | 165.34(5) | 165.33(6) | 165.26(8) |
| Cocrystal | (tftib)(AsPh$_3$) @103 K | (tftib)(AsPh$_3$) @153 K | (tftib)(AsPh$_3$) @203 K | (tftib)(AsPh$_3$) @253 K |
| $T$ (K) | 103.0(1) | 153.0(1) | 203.0(1) | 253.0(1) |
| $d_{XB,exp}$ (Å) | 3.4211(3) | 3.4412(3) | 3.4631(4) | 3.4886(7) |
| $R_{XB}$ | 0.893 | 0.898 | 0.904 | 0.911 |
| $\angle_{C-I···E}$ (°) | 166.02(5) | 165.89(6) | 165.81(7) | 165.73(9) |
| Cocrystal | (tftib)(SbPh$_3$) @103 K | (tftib)(SbPh$_3$) @153 K | (tftib)(SbPh$_3$) @203 K | (tftib)(SbPh$_3$) @253 K |
| $T$ (K) | 103.0(1) | 153.0(1) | 203.0(1) | 253.0(1) |
| $d_{XB,exp}$ (Å) | 3.5747(3) | 3.5873(4) | 3.6125(3) | 3.6307(4) |
| $R_{XB}$ | 0.885 | 0.888 | 0.894 | 0.899 |
| $\angle_{C-I···E}$ (°) | 168.28(6) | 168.14(6) | 168.00(8) | 167.79(10) |

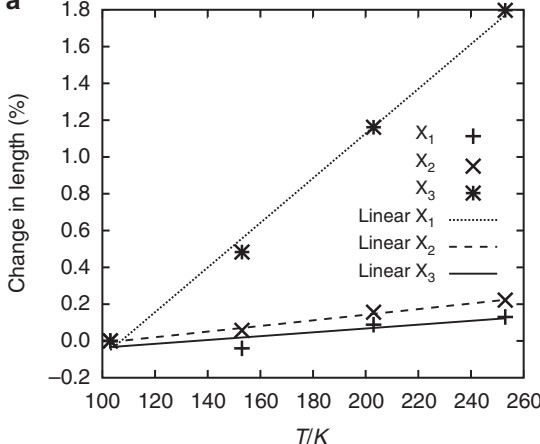

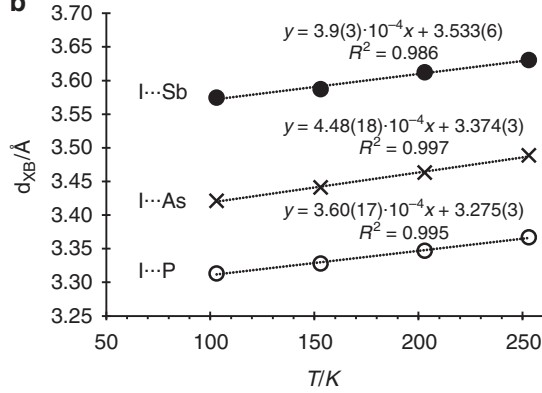

**Fig. 5** Selected thermal properties of the herein presented cocrystals. **a** Temperature dependence of the principal axes lengths reveals colossal thermal expansion for (**tftib**)(**SbPh$_3$**). For corresponding data on (**tftib**)(**PPh$_3$**) and (**tftib**)(**AsPh$_3$**) see Supplementary Figs. 27-29. **b** Temperature-dependent change in $d_{XB}$ for (**tftib**)(**PPh$_3$**) (bottom), (**tftib**)(**AsPh$_3$**) (middle), and (**tftib**)(**SbPh$_3$**) (top)

triphenylbismuth (**BiPh$_3$**) were purchased from Sigma-Aldrich, while 1,3,5-tri-fluoro-2,4,6-triiodobenzene (**tftib**) was purchased from Ark Pharm.

**Mechanochemical experiments.** Mechanochemical experiments were conducted on a Retsch MM200 mill operating at 25 Hz frequency using a 14 mL polytetra-fluoroethylene (PTFE) jar along with a zirconia ball (9.5 mm diameter and ~3 g weight).

**Single crystal X-ray diffraction.** Single crystal X-ray diffraction data for (**tftib**)(**PPh$_3$**), (**tftib**)(**AsPh$_3$**), and (**tftib**)(**SbPh$_3$**) were collected at 103, 153, 203, and 253 K on a Bruker D8 Venture dual-source diffractometer equipped with a PHOTON 100 detector and an Oxford Cryostream 800 cooling system, using mirror-monochromatized MoK$\alpha$ radiation ($\lambda = 0.71073$ Å) from a microfocus source. Data were collected in a series of $\varphi$- and $\omega$-scans. APEX3 software was used for data collection, integration and reduction[61]. Numerical absorption corrections were applied using SADABS-2016/2[62]. The structures were solved by dual-space iterative methods using SHELXT[63] and refined by full-matrix least-squares on $F^2$ using all data with SHELXL[64] within the OLEX2[65] and/or WinGX[66] environment. In all cases, some reflections were found to have been obscured by the beam stop and were omitted from the refinement. Extinction correction was applied in all cases. Hydrogen atoms were placed in calculated positions and treated as riding on the parent carbon atoms with $U_{iso}$(H) = 1.2 $U_{eq}$(C). Crystal structure figures were generated using Mercury[67] and POV-Ray[68]. The X-ray crystallographic coordinates for structures reported in this study have been deposited at the Cambridge Crystallographic Data Centre (CCDC), under deposition numbers 1850430-1850441. These data can be obtained free of charge from The Cambridge Crystallographic Data Centre via www.ccdc.cam.ac.uk/structures.

**Powder X-ray diffraction (PXRD).** Powder X-ray diffraction experiments were performed on a Bruker D8 Advance diffractometer with CuK$\alpha$ ($\lambda = 1.54184$ Å) radiation source operating at 40 mA and 40 kV, equipped with a Lynxeye XE linear position sensitive detector, or a Bruker D2 Phaser diffractometer with CuK$\alpha$ ($\lambda = 1.54184$ Å) radiation source operating at 10 mA and 30 kV, equipped with a Lynxeye linear position sensitive detector, in the $2\theta$ range of 4/5−40°.

**Thermal analysis.** Differential scanning calorimetry (DSC) measurements were performed on a Mettler-Toledo DSC823$^e$ module in sealed aluminum pans (40 μL) with three pinholes in the lid, heated in a stream of nitrogen (150 mL min$^{-1}$) at a heating rate of 10 °C min$^{-1}$. Thermogravimetric analysis (TGA) measurements were performed on a Mettler-Toledo TGA/SDTA 851$^e$ module in sealed aluminum pans (40 μL) with three pinholes in the lid, heated in a stream of nitrogen (150 mL min$^{-1}$) at a heating rate of 10 °C min$^{-1}$. Data collection and analysis were performed using the program package STAR$^e$ Software 15.00.

**Infrared spectroscopy.** Fourier-transform infrared attenuated total reflectance (FTIR-ATR) measurements were performed on a Bruker VERTEX 70 instrument equipped with a single-reflection diamond crystal Platinum ATR unit.

**In situ Raman spectroscopy.** All Raman experiments were performed using a RamanRxn1™ analyzer by Kaiser Optical Systems Inc. operating with a 785 nm laser. Spectra were dark- and intensity-corrected using the Holograms® software package before being processed in MATLAB, where spectra were smoothed using a Savitsky-Golay filter, background-corrected with the Sonneveld-Visser algorithm[69], and normalized via standard normal variate. In situ Raman monitoring experiments were performed with the same loadings as described in the preparation of cocrystals. Reactions were performed using a Retsch MM400 mill operating at 25 Hz equipped with a custom designed sapphire jar with a volume of 8.5 mL and loaded with one 2 grams zirconia ball. All measurements were run for 30 min with a spectrum acquisition every 5 s.

**Cocrystal synthesis and single crystal growth.** The cocrystal (**tftib**)(**PPh$_3$**) was prepared by milling a mixture of **tftib** (132.1 mg, 0.259 mmol) and **PPh$_3$** (67.9 mg, 0.259 mmol), either neat or along with 40.0 μL of ethanol for 20 min. Single crystals were obtained by dissolving a mixture of **tftib** (58.3 mg, 0.114 mmol) and **PPh$_3$**

(30.0 mg, 0.114 mmol) in 1 mL of acetonitrile and leaving the solution to cool and evaporate at room temperature for 1 day.

The cocrystal (**tftib**)(**AsPh₃**) was prepared by milling a mixture of **tftib** (124.9 mg, 0.245 mmol) and **AsPh₃** (75.1 mg, 0.245 mmol), either neat or along with 40.0 µL of ethanol for 20 min. Single crystals were obtained by dissolving a mixture of **tftib** (49.9 mg, 0.098 mmol) and **AsPh₃** (30.0 mg, 0.098 mmol) in 1 mL of acetonitrile and leaving the solution to cool and evaporate at room temperature for 1 day.

The cocrystal (**tftib**)(**SbPh₃**) was prepared by milling a mixture of **tftib** (118.2 mg, 0.232 mmol) and **SbPh₃** (81.8 mg, 0.232 mmol), either neat or along with 40.0 µL of ethanol for 20 min. Single crystals were obtained by dissolving a mixture of **tftib** (43.3 mg, 0.085 mmol) and **SbPh₃** (30.0 mg, 0.085 mmol) in 1 mL of acetonitrile and leaving the solution to cool and evaporate at room temperature for 1 day.

**Computational methods**. DFT calculations (both periodic and molecular) were performed using the program CRYSTAL17[37]. The calculations utilized range-separated hybrid ωB97X functional, and electronic wavefunctions were modeled with a POB-TZVP basis set[38–40], specifically modified from a standard TZVP basis set for use in periodic calculations. The basis sets for heavy elements (Sb, I) were supplemented by effective core potentials (ECPs), screening 28 core electrons ($n = 1, 2$, and 3 shells). In the case of Bi atom, a different basis set with 60-electron ECP was used[41], since the POB-TZVP basis set had not been parameterized for the elements in Period 6.

Periodic DFT geometry optimizations were performed for the experimentally determined cocrystal structures of (**tftib**)(**EPh₃**) (E = P, As, Sb), putative (**tftib**)(**BiPh₃**) structure, as well as the starting materials. For the starting materials, the crystal structures were obtained from CSD: **tftib** (UCEPEY), **PPh₃** (PTRPHE02), **AsPh₃** (ZZZEIG01), **SbPh₃** (ZZEHA01), and **BiPh₃** (BITRPH02). Geometry optimization of all crystal structures involved relaxation of atom coordinates and unit cell parameters, subject to the constraints of the corresponding space groups. The following convergence criteria were used: maximum force $4.5 \cdot 10^{-4}$ Ha Bohr$^{-1}$, RMS force $3.0 \cdot 10^{-4}$ Ha Bohr$^{-1}$, maximum atom displacement $1.8 \cdot 10^{-3}$ Bohr, RMS atom displacement $1.2 \cdot 10^{-3}$ Bohr. The energies of all crystal structures were corrected for basis set superposition error (BSSE) using counterpoise method with ghost atoms located up to 5 Å away from the reference molecule. The final electronic energies for the optimized crystal structures, as well as calculated cocrystal formation energies are shown in the Supplementary Table 4. Supplementary Data 1–9 contain coordinates for all DFT-optimized solid-state structures in CIF format (see Supplementary Note 1).

We have validated the choice of range-separated ωB97X functional by comparing geometries of DFT-optimized and experimental cocrystal structures. The calculated halogen bond lengths all fall within 0.1 Å of their experimental values, which accounts for less than 3% relative error (Supplementary Table 8). The deviation of calculated halogen bond angles, $\angle_{\mathrm{C-I\cdots E}}$, is always below 6°, i.e., below 4% from their experimental values. Periodic DFT calculations provide superior accuracy, and our calculated halogen bond parameters are in line with the general accuracy of non-covalent interactions achievable with periodic DFT methods[45]. In contrast, gas-phase simulations of halogen-bonded complexes neglect the effects of crystal packing, resulting in up to 0.2 Å deviations from experimental solid-state halogen bond lengths[46].

The overall accuracy of periodic DFT structure optimizations can be evaluated by comparing the optimized unit cell parameters with their experimental values (Supplementary Table 9). The optimized unit cell lengths are all 2-5% shorter than their experimental values. This shortening of the unit cell axes results in 7–8% lower calculated unit cell volumes for all three cocrystals. The differences in unit cell volumes can be partially attributed to the temperature effects, since DFT calculations correspond to 0 K temperature, while experimental unit cell parameters were measured at 103 K. In terms of unit cell angles, the deviation from experimental values is in the 0.5–4.8° range (0.5–6.8% relative error). The overall agreement of unit cell parameters is within the limits of expected accuracy of periodic DFT calculations for organic molecular crystals, which are held together only by non-covalent interactions[47,48]. Our results confirm that ωB97X range-separated functional provides an adequate description of supramolecular interactions present in the cocrystal structures.

Molecular calculations were performed for the halogen-bonded dimers extracted from the optimized cocrystal structures. Two sets of dimerization energies were calculated, one for the dimer with the geometry found in crystal lattice, another for the dimer optimized in the gas-phase geometry. Dimer dissociation energies were BSSE-corrected using counterpoise method and are summarized in Supplementary Tables 5 and 6. Supplementary Data 10–14 contain geometries for all optimized gas-phase dimers in XYZ format (see Supplementary Note 2).

Electron density distributions of individual molecules were calculated using Gaussian 16 program, and the ESP surfaces were plotted with GaussView 6[43,44]. The calculations utilized ωB97X functional and Def2-TZVP molecular basis set combined with 28-electron (for Sb and I) and 60-electron (for Bi) ECPs. The ESP maps were plotted on a 0.002 a.u. total electron density isosurfaces.

Spin-orbit coupling (SOC) calculation was performed for the halogen-bonded dimer (**BiPh₃**)(**tftib**) in the geometry extracted from the optimized crystal structure. The calculation was performed using the program ADF version 2018.101[53,54], employing the ωB97X functional with double-zeta (DZP) all-electron basis set[70]. Relativistic effects were simulated with a spin-orbit zeroth order regular approximation (ZORA) method[71]. In addition, a standard DFT calculation, uncorrected for relativistic effects, was also performed to assess the contribution of SOC to the halogen bond interaction energy.

## Data availability

All relevant data, including powder X-ray diffraction (PXRD), Fourier-transform infrared attenuated total reflectance (FTIR-ATR) and Raman spectroscopy, as well as differential scanning calorimetry and thermogravimetric data that support the findings of this study are available on request from the authors. The X-ray crystallographic coordinates for structures reported in this study have been deposited at the Cambridge Crystallographic Data Centre (CCDC), under deposition numbers 1850430-1850441. These data can be obtained free of charge from The Cambridge Crystallographic Data Centre via www.ccdc.cam.ac.uk/data_request/cif. The authors declare that all other data supporting the findings of this study are available within the paper and its supplementary information files.

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

## Acknowledgements

We thank the Croatian Science Foundation (IP-2014-09-7367), NSERC Discovery Grant (RGPIN-2017-06467) and E. W. R. Steacie Memorial Fellowship (SMFSU 507347-17). A.J.M. thanks the University of Birmingham for research funding through the Birmingham Fellowship scheme. F.T. thanks the Banting Postdoctoral Fellowship from the Government of Canada. The research was enabled in part by support of Calcul Québec (http://www. calculquebec.ca) and Compute Canada (www.computecanada.ca). McGill University is

acknowledged for the Gaussian 16 licence. We also thank SHARCNET (https://www.sharcnet.ca) for the ADF software licence and access to the Graham supercomputer.

## Author contributions

K.L., F.T., S.C., and T.F. performed experimental cocrystal synthesis and characterization. Theoretical calculations were performed by M. A. with assistance from A.J.M. In situ Raman measurements were performed by P.A.J. using the milling equipment specifically developed by C.W.N. The manuscript was written by T.F. and D.C. with contributions from all authors.

## Additional information

**Competing interests:** The authors declare no competing interests.

