## [Transparent Peer Review File · Nature Communications]

Reviewers' comments:

Reviewer #1 (Remarks to the Author):

The manuscript reports the first observation of halogen-bond (XB) cocrystals involving neutral phosphorus, arsenic and antimony acceptors. Currently, halogen bonding is a hot topic. The strategy used is based on a correlation between experimental results and DFT calculations. This interesting work fits very well the "Nature Communications" journal. However, I have some comments that need to be addressed before publication.

As a theoretical chemist, I do not feel able to assess the quality of the experimental part and I focused on the computational aspects.

Major comments:

1. the presentation of the computational methodology must be improved since some given information can be (i) incorrect or (ii) misleading:

(i) - lines 374-375 "In the case of Bi atom, a different basis set with 60-electron ECP was used,³⁸" but the article with reference 38 give no information on Bi ECP and basis set !

- lines 401-402 "The calculations utilized ω B97X functional and Def2-TZVP molecular basis set. combined with 28-electron ECPs for Sb, Bi and I atoms." It is incorrect for Bi since the def2-TZVP basis set is built with a 60-electron ECP for Bi.

- lines 154-156 "Periodic and gas-phase geometry optimizations were performed using CRYSTAL17³⁴ with all-electron basis sets.³⁵⁻³⁷ Effective core potentials (ECPs)³⁸ were used to account for relativistic effects in heavy atoms (Sb, I, Bi)." The two sentences are just contradictory, the text should be revised.

(ii) in introduction to the presentation of the theoretical results, in the main text section, the choice to use for this work the ω B97X functional is justified a priori on the basis of studies from the literature: lines 149-151 "we used the range-separated ω B97X functional²⁹ which is expected to yield a more accurate description of XB contacts and crystal structure as a whole.^{30,31}" However, neither the performances of the ω B97X one nor those of the range-separated functionals are discussed in the article with reference 30. Furthermore, reference 31 is about works focussed on halogen-bond interactions with delocalized π systems, which is not the case in the current study.

A more convincing approach (than wrongly citing the literature) to validate the computational methodology would be to compare the available theoretical results to the experimental data (for instance the XB geometrical parameters from the calculated and XRD cocrystals). A short and discussed comparison (in Supporting Information?) would be desirable.

2. In Table 2 and Table S4, the authors present "calculated cocrystal formation enthalpies ($\Delta H_f, calc$)", and "Dimer enthalpy of formation" in Table S5 and Table S6. However, the reported values are calculated from electronic energies. Hence, these values are not enthalpies but rather interaction energies.

3. A strong case is made on the positive value of the calculated "formation enthalpy" for the (tftib)(BiPh₃) cocrystal in order to justify the non-observation of this cocrystal, c.f. discussion lines 222-229. However, the calculated value, +1.27 kJ mol⁻¹, is rather small and actually can be found negative at a higher level of theory. Indeed, the spin-dependent relativistic effects are disregarded for heavy atoms while it is stated that the computations "account for relativistic effects" (line 156). Bi is a 6p element and it was previously shown that the spin-orbit coupling can modifies the interaction energies for halogen bonds involving 6p elements up to 35% (P. Matczak, Mol. Phys., 2018, p 338-350) and 9 kJ mol⁻¹ (N. Galland et al., New J. Chem., 2018, p 10510-10517). Hence, the influence of the spin-orbit coupling on the energies reported in Table 2 for the BiPh₃ acceptor must be checked. At least, the authors can evaluate the spin-orbit coupling effects on the interaction

energy of the (tftib)(BiPh₃) dimer in gas phase, which will provide indications on how other energies can be modified and how the conclusions regarding the (tftib)(BiPh₃) cocrystal can be affected.

Minor comments:

1. The topic of this paper is halogen bonding, so the authors should at least provide a definition of the halogen bond interaction in the main text.

2. Line 180-184, the authors write "This trend is explained by a combination of several factors: drop in ESP due to decreasing electronegativity of the pyramidally-bonded pnictogen atom in the order PPh₃ > AsPh₃ > SbPh₃ > BiPh₃, along with the delocalization of the lone electron pair and increased steric bulk surrounding the more planar nitrogen atom in NPh₃ (Figure 4)." The different "factors" should be more highlighted for improving the readability of the text.

3. Regarding the NPh₃ acceptor, the argument that the nitrogen lone pair is delocalized is used several times by the authors (lines 182, 201). However, this delocalization is not proved and it can be noticed that the phenyl rings are not coplanar neither between them nor with the plane formed by the nitrogen and the bound carbon atoms.

4. Line 49, the authors should add some recent references of XB cocrystals involving nitrogen.

5. There are some typo throughout the manuscript:

- Line 167, "Figure 3" should be "Figure 4".
- Line 401, the point "." after the "set" word should be removed or replaced by a comma.
- Line 517, please remove the large spacing between "quality" and "of".
- Line 522, "moelcular" should be "molecular".

Reviewer #2 (Remarks to the Author):

This is a well-written paper that expands available halogen bonding acceptors deeper into the pnictogen group. The work is done well, and a thorough analysis of the structures, structural dynamics, thermodynamics and thermal behavior of the three co-crystals have been reported. This work will be of great interest to the halogen-bonding community, as well as the chemical community in general. The variety and uniqueness of some of the techniques used will be instructive for many.

There are a few comments that should be considered before acceptance of the paper.

- 1) While the authors do briefly discuss this later in the paper, it should be pointed out as fairly obvious that the triphenylamine X-acceptor will be too sterically protected to form halogen bonds.
- 2) The authors also briefly mention the role of phenyl interactions in the packing of the molecules, but have they looked closely for the presence of phenyl embraces, as described in many elegant papers by Ian Dance.
- 3) The term "colossal" to describe the thermal expansion of the c-axis in the Ph₃Sb co-crystal seems a little over the top. The difference in rate of expansion of this axis versus the others is interesting, but the lack of expansion along the other axes seems more surprising. Although expansion along the longest crystallographic axis is probably to be expected. Analysis of the DSC for this compound doesn't show any obvious signs of a phase change, but this might be something to consider, especially given the greater expansion of this compound than the others. Rearrangement of phenyl interactions might be involved in a subtle phase change that might be of low enough energy to be difficult to determine by DSC.

Reviewer #3 (Remarks to the Author):

Crystallographic review.

Accept for publication after amendment to data analysis.

The manuscript submission includes twelve crystal data sets for three individual crystals each measured at four different temperatures. The work discusses the iso-structural nature of the three crystals and reports in detail the nature of the halogen-pnictogen intermolecular bonds. The twelve crystal structures have been refined well and are of sufficient quality to support the scientific discussion in the paper. The authors have included checkCIF reports for all structures and responded satisfactorily to relevant alerts generated therein.

However, before publication can be recommended an issue needs to be addressed in the analysis of intermolecular distances and angles. The values reported in the manuscript appear to have been generated from the CIF matrix alone. These values and their uncertainties should be calculated by inclusion of an RTAB instruction in the refinement input file. This will refine the uncertainties against the full matrix and output the value at the end of the list file.

e.g. for tpa_tftib_103K.cif

Using the geometry tool in Olex2 gives the following values and uncertainties

As1a ...I1s = 3.4213(4) Å

As1a-I1s-C1s = 166.02(5)°

Whereas using the command

```
RTAB dist As1a I1s
```

```
RTAB angl C1s I1s As1a
```

And refining with SHELXL gives

```
"Distance DIST
```

```
3.4211 (0.0003) I1S - As1A
```

```
Angle ANGL
```

```
166.02 ( 0.05) As1A - I1S - C1S"
```

New values and uncertainties should be determined by full matrix refinement for the halogen bond distances and angles which are central to the discussion in this work. Tables of bond lengths and angles should be updated. Resubmission of new crystallographic files is not necessary.

Reviewer #4 (Remarks to the Author):

The paper by Friscic and Cincic, et al., reports the structural characterization of halogen-bonded cocrystals involving P, As, and Sb as acceptors. Despite the structures with iodofluoroaromatics are new and interesting, I cannot recommend acceptance of this paper in Nat. Comm.

The reported examples are not the first examples of halogen-bonded cocrystals involving P, As, and Sb, as the authors state in various parts of the paper. In fact, there are structures of Ph₃P, Ph₃As, and Ph₃Sb with I₂, which are identical to the cocrystal between DABCO and I₂. These previous reports hamper quite much the novelty of this paper.

Furthermore, the new cocrystals do not display any particular or unexpected behavior from the application point of you. Therefore, what is left is a good article about solid-state characterization of new halogen-bonded co-crystal, which should be published in a journal specialized in crystallography.

RESPONSES TO REMARKS OF REVIEWER #1:

The manuscript reports the first observation of halogen-bond (XB) cocrystals involving neutral phosphorus, arsenic and antimony acceptors. Currently, halogen bonding is a hot topic. The strategy used is based on a correlation between experimental results and DFT calculations. This interesting work fits very well the "Nature Communications" journal. However, I have some comments that need to be addressed before publication.

As a theoretical chemist, I do not feel able to assess the quality of the experimental part and I focused on the computational aspects.

Major comments:

Comment 1. the presentation of the computational methodology must be improved since some given information can be (i) incorrect or (ii) misleading:

(i) - lines 374-375 "In the case of Bi atom, a different basis set with 60-electron ECP was used,³⁸" but the article with reference 38 give no information on Bi ECP and basis set!

Response: We are grateful to the Referee for noting this issue – we have accidentally mis-numbered the references. We have indeed used a 60-electron ECP set for bismuth, as well as 28-

electron ECP sets for antimony and iodine. We have now re-ordered the references and corrected this mistake. We have also provided additional clarifications in the text.

Comment: - lines 401-402 "The calculations utilized ω B97X functional and Def2-TZVP molecular basis set. combined with 28-electron ECPs for Sb, Bi and I atoms." It is incorrect for Bi since the def2-TZVP basis set is built with a 60-electron ECP for Bi.

Response: Again, as explained above, we are grateful to the Referee for noting this inadvertent mistake in composing the text – we have now clarified in the manuscript that 28-electron ECPs were used for Sb and I only, and a 60-electron ECP was employed for bismuth.

Comment: - lines 154-156 "Periodic and gas-phase geometry optimizations were performed using CRYSTAL17³⁴ with all-electron basis sets.³⁵⁻³⁷ Effective core potentials (ECPs)³⁸ were used to account for relativistic effects in heavy atoms (Sb, I, Bi)." The two sentences are just contradictory, the text should be revised.

Response: Again, we thank the Referee for noting this mistake in our presentation. We have now revised this section to clarify that all-electron basis sets were used for all elements up to and including arsenic, while 28-electron ECPs were used for antimony, iodine, and a 60-electron ECP set was used for bismuth.

Comment: (ii) in introduction to the presentation of the theoretical results, in the main text section, the choice to use for this work the ω B97X functional is justified a priori on the basis of studies from the literature: lines 149-151 "we used the range-separated ω B97X functional²⁹ which is expected to yield a more accurate description of XB contacts and crystal structure as a whole.^{30,31}" However, neither the performances of the ω B97X one nor those of the range-separated functionals are discussed in the article with reference 30. Furthermore, reference 31 is about works focussed on halogen-bond interactions with delocalized π systems, which is not the case in the current study. A more convincing approach (than wrongly citing the literature) to validate the computational methodology would be to compare the available theoretical results to the experimental data (for instance the XB geometrical parameters from the calculated and XRD cocrystals). A short and discussed comparison (in Supporting Information?) would be desirable.

Response: We thank the Referee for the meticulous reading of the manuscript and agree with them. The problematic references were placed into the manuscript with the desire to highlight the recent work in theoretical modelling of halogen bonds in general. Obviously, we have not presented this clearly. We have now revised the text so that it is clear where we are commenting on general aspects of halogen bond calculations, and where we are referencing a specific functional.

We have also heeded the Referee's suggestion and examined the comparison of experimental and theoretical halogen bond geometries for solid-state structures, noting in the text that the agreement is highly satisfactory. The data is provided in a new table in the Supplementary Information (Supplementary Table S8). As, to the best of our knowledge, there have not yet been any benchmark comparisons of halogen bond geometries measured in a crystal and calculated using periodic DFT, we have provided a reference illustrating how our results fit into the general expectations from periodic DFT modelling of supramolecular interactions in the solid state (reference 44), as well as a reference to a recent benchmark study that was comparing halogen bond geometries measured in the solid state to those calculated in the gas phase (reference 45).

Finally, to further ascertain the accuracy and quality of our periodic DFT modelling, we have also provided an additional table in the Supplementary Information (Supplementary Table S9) illustrating the differences between unit cell parameters and volumes measured at 103 K and optimized using periodic DFT.

As the Supplementary Information is not intended to contain text, we have provided the requested discussion in different parts of the manuscript text and the experimental section.

Comment: 2. In Table 2 and Table S4, the authors present "calculated cocrystal formation enthalpies ($\Delta H_{f,calc}$)", and "Dimer enthalpy of formation" in Table S5 and Table S6. However, the reported values are calculated from electronic energies. Hence, these values are not enthalpies but rather interaction energies.

Response: We thank the Referee for noting this, we have revised the manuscript, replacing formation enthalpies with "formation energies", and the symbol " H " with " E "

Comment: 3. A strong case is made on the positive value of the calculated "formation enthalpy" for the (tftib)(BiPh₃) cocrystal in order to justify the non-observation of this cocrystal, c.f. discussion lines 222-229. However, the calculated value, +1.27 kJ mol⁻¹, is rather small and actually can be found negative at a higher level of theory. Indeed, the spin-dependent relativistic effects are disregarded for heavy atoms while it is stated that the computations "account for relativistic effects" (line 156). Bi is a 6p element and it was previously shown that the spin-orbit coupling can modify the interaction energies for halogen bonds involving 6p elements up to 35% (P. Matczak, Mol. Phys., 2018, p 338-350) and 9 kJ mol⁻¹ (N. Galland et al., New J. Chem., 2018, p 10510-10517). Hence, the influence of the spin-orbit coupling on the energies reported in Table 2 for the BiPh₃ acceptor must be checked. At least, the authors can evaluate the spin-orbit coupling effects on the interaction energy of the (tftib)(BiPh₃) dimer in gas phase, which will provide indications on how other energies can be modified and how the conclusions regarding the (tftib)(BiPh₃) cocrystal can be affected.

Response: We thank the Referee for the advice which we have now followed, and evaluated the dissociation energy for the isolated BiPh₃-tftib complex taking into account the spin-orbit coupling. Our results, which are mentioned in the text and also provided in the Supplementary (Supplementary Table S6), show that including spin-orbit coupling into our calculations does not noticeably affect their outcome.

Minor comments:

Comment: 1. The topic of this paper is halogen bonding, so the authors should at least provide a definition of the halogen bond interaction in the main text.

Response: We completely agree with the Referee and have now provided a brief comment along with a relevant reference (new reference 1) in the manuscript introduction.

Comment 2. Line 180-184, the authors write "This trend is explained by a combination of several factors: drop in ESP due to decreasing electronegativity of the pyramidally-bonded pnictogen atom in the order PPh₃ > AsPh₃ > SbPh₃ > BiPh₃, along with the delocalization of the lone electron pair and increased steric bulk surrounding the more planar nitrogen atom in NPh₃ (Figure 4)." The different "factors" should be more highlighted for improving the readability of the text.

Response: Following the Referee's comments, we have now provided a more systematic, point-by-point description of each of these factors, while replacing the discussion of delocalization with the computational observation of a reduced ESP on the nitrogen atom.

Comment: 3. Regarding the NPh_3 acceptor, the argument that the nitrogen lone pair is delocalized is used several times by the authors (lines 182, 201). However, this delocalization is not proved and it can be noticed that the phenyl rings are not coplanar neither between them nor with the plane formed by the nitrogen and the bound carbon atoms.

Response: Delocalisation of the nitrogen lone pair in NPh_3 is a well-known phenomenon that has been extensively studied using theoretical and experimental techniques, and has also previously been used to explain the poor basicity of the nitrogen atom in this molecule compared to, for example, aliphatic amines. To this effect, we have now provided a reference addressing the nitrogen lone pair delocalization with the π -systems of the attached phenyl groups (reference 48). Further references can be provided, if needed (e.g. Munshi *et al. Phys. Chem. Chem. Phys.* **19**, 19881-19889 (2017)), but these would exceed the limit of 70 references.

Comment: 4. Line 49, the authors should add some recent references of XB cocrystals involving nitrogen.

Response: Following the Referee's advice we have now included several recent references from the literature (references 13-15). At the same time, we also had to remove some of previously included references due to the required 70 references limit.

Comment: 5. There are some typo throughout the manuscript:

- Line 167, "Figure 3" should be "Figure 4".
- Line 401, the point "." after the "set" word should be removed or replaced by a comma.
- Line 517, please remove the large spacing between "quality" and "of".
- Line 522, "moelcular" should be "molecular".

Response: We thank the referee for noticing these typos, they have been corrected.

RESPONSES TO REMARKS OF REVIEWER #2:

This is a well-written paper that expands available halogen bonding acceptors deeper into the pnictogen group. The work is done well, and a thorough analysis of the structures, structural dynamics, thermodynamics and thermal behavior of the three co-crystals have been reported. This work will be of great interest to the halogen-bonding community, as well as the chemical community in general. The variety and uniqueness of some of the techniques used will be instructive for many.

There are a few comments that should be considered before acceptance of the paper.

Comment: 1) While the authors do briefly discuss this later in the paper, it should be pointed out as fairly obvious that the triphenylamine X-acceptor will be too sterically protected to form halogen bonds.

Response: We have now noted in the manuscript that NPh_3 exhibits a different bonding geometry around the nitrogen, compared to its pyramidal P, As, Sb and Bi derivatives, and that this leads to more steric congestion.

Comment: 2) The authors also briefly mention the role of phenyl interactions in the packing of

the molecules, but have they looked closely for the presence of phenyl embraces, as described in many elegant papers by Ian Dance.

Response: We thank the Referee for the comment, and we have now added a note in the manuscript on the observation of phenyl embraces, as well as an additional image illustrating these motifs in Supplementary Information (new Supplementary Figure S2). We have also included a relevant reference to the work of the Dance group on this topic (reference 25).

Comment: 3) The term "colossal" to describe the thermal expansion of the c-axis in the Ph₃Sb co-crystal seems a little over the top. The difference in rate of expansion of this axis versus the others is interesting, but the lack of expansion along the other axes seems more surprising. Although expansion along the longest crystallographic axis is probably to be expected. Analysis of the DSC for this compound doesn't show any obvious signs of a phase change, but this might be something to consider, especially given the greater expansion of this compound than the others. Rearrangement of phenyl interactions might be involved in a subtle phase change that might be of low enough energy to be difficult to determine by DSC.

Response: We use this term not to impress but to follow the existing notation in the field. The term "colossal" was introduced as such by Goodwin et al. a decade ago (ref. 56 in the manuscript, *i.e.* Goodwin, A. L. *et al. Science* **319**, 794–797 (2008)) to describe thermal expansion where the magnitude of the linear expansion coefficient is higher than 100 MK⁻¹ (*i.e.* it is either > 100 MK⁻¹ or < -100 MK⁻¹). The term "colossal" has been commonly used for that purpose in the literature ever since.

As the Referee noted, we have not been able to observe any evidence of a phase transformation using DSC measurements. However, the thermal expansion data that the Reviewer is referring to is based on single crystal X-ray diffraction at different temperatures, which would have certainly demonstrated a phase transition in the relevant temperature range.

RESPONSES TO REMARKS OF REVIEWER #3:

Crystallographic review.

Accept for publication after amendment to data analysis.

The manuscript submission includes twelve crystal data sets for three individual crystals each measured at four different temperatures. The work discusses the iso-structural nature of the three crystals and reports in detail the nature of the halogen-pnictogen intermolecular bonds. The twelve crystal structures have been refined well and are of sufficient quality to support the scientific discussion in the paper. The authors have included checkCIF reports for all structures and responded satisfactorily to relevant alerts generated therein.

However, before publication can be recommended an issue needs to be addressed in the analysis of intermolecular distances and angles. The values reported in the manuscript appear to have been generated from the CIF matrix alone. These values and their uncertainties should be calculated by inclusion of an RTAB instruction in the refinement input file. This will refine the uncertainties against the full matrix and output the value at the end of the list file.

e.g. for tpa_tftib_103K.cif

Using the geometry tool in Olex2 gives the following values and uncertainties

As1a ...I1s = 3.4213(4) Å

As1a-I1s-C1s = 166.02(5)°

Whereas using the command
RTAB dist As1a I1s
RTAB angl C1s I1s As1a
And refining with SHELXL gives
“Distance DIST 3.4211 (0.0003) I1S - As1A
Angle ANGL 166.02 (0.05) As1A - I1S - C1S”

Comment: New values and uncertainties should be determined by full matrix refinement for the halogen bond distances and angles which are central to the discussion in this work. Tables of bond lengths and angles should be updated. Resubmission of new crystallographic files is not necessary.

Response: We thank the Referee for reviewing our work. We have now provided updated values and uncertainties of halogen bonds, calculated as recommended by the Referee.

RESPONSES TO REMARKS OF REVIEWER #4:

Comment: The paper by Friscic and Cincic, et al., reports the structural characterization of halogen-bonded cocrystals involving P, As, and Sb as acceptors. Despite the structures with iodofluoroaromatics are new and interesting, I cannot recommend acceptance of this paper in Nat. Comm. The reported examples are not the first examples of halogen-bonded cocrystals involving P, As, and Sb, as the authors state in various parts of the paper. In fact, there are structures of Ph₃P, Ph₃As, and Ph₃Sb with I₂, which are identical to the cocrystal between DABCO and I₂. These previous reports hamper quite much the novelty of this paper.

Response: The systems mentioned by the referee have already been explicitly noted in our paper, as examples of covalent adducts (references 24 and 26). The pnictogen-iodine distances in these adducts have already been provided in the manuscript, each time with a comment that they are an excellent match to those expected for covalent bond formation. We have now strengthened each of those comments by also explicitly listing the covalent radii of P, As and I in the paper (as reported in reference 23). Consequently, the previously described systems cannot be regarded as cocrystals, but rather as covalent molecules, as already noted in the manuscript.

Comment: Furthermore, the new cocrystals do not display any particular or unexpected behavior from the application point of you.

Response: The prepared cocrystals display very different behavior from the covalent molecules noted by the referee (and already cited in references 24 and 26). The cocrystals can readily be dissolved and re-formed from different solvents. In contrast, the covalent adducts noted by the referee readily rearrange and dissociate in solution to form halonium salts. This is a significant difference in behavior and has been highlighted in several parts of the manuscript.

Comment: Therefore, what is left is a good article about solid-state characterization of new halogen-bonded co-crystal, which should be published in a journal specialized in crystallography.

Response: Based on the comments of other Referees, and our responses to this Referee's comments above, we respectfully disagree and believe we have made a strong enough case that this type of halogen-bonding interactions has not previously been observed, that it provides more liberty in materials synthesis (by avoiding re-arrangement and dissociation leading to halonium salts), and that it can lead to exciting new properties (e.g. colossal thermal expansion).

REVIEWERS' COMMENTS:

Reviewer #1 (Remarks to the Author):

Editorial note: this reviewer provided no further comments to the authors.

Reviewer #2 (Remarks to the Author):

The changes made have improved the manuscript, and it can now be accepted "as is".